# The Use of Brain Natriuretic Peptide in the Evaluation of Heart Failure in Geriatric Patients

**DOI:** 10.3390/diagnostics13091512

**Published:** 2023-04-23

**Authors:** Mihai Marinescu, Violeta Diana Oprea, Aurel Nechita, Dana Tutunaru, Luiza-Camelia Nechita, Aurelia Romila

**Affiliations:** 1Faculty of Medicine and Pharmacy, “Dunărea de Jos” University in Galați, 800216 Galați, Romania; 2“St. Apostle Andrei” Clinical Emergency County Hospital, 800578 Galați, Romania; 3“St. Ioan” Emergency Clinical Hospital for Children, 800487 Galați, Romania

**Keywords:** heart failure in geriatric population, natriuretic peptides in heart failure, BNP, NTproBNP

## Abstract

Heart failure is one of the main morbidity and mortality factors in the general population and especially in elderly patients. Thus, at the European level, the prevalence of heart failure is 1% in people under 55 years of age but increases to over 10% in people over 70 years of age. The particularities of the elderly patient, which make the management of heart failure difficult, are the presence of comorbidities, frailty, cognitive impairment and polypharmacy. However, elderly patients are under-represented in clinical trials on the diagnosis and treatment of heart failure. The need for complementary methods (biomarkers) for differential and early diagnosis of heart failure is becoming more and more evident, even in its subclinical stages. These methods need to have increased specificity and sensitivity and be widely available. Natriuretic peptides, in particular B-type natriuretic peptide (BNP) and its fraction NTproBNP, have gained an increasingly important role in the screening, diagnosis and treatment of heart failure in recent years.

## 1. Introduction

Heart failure is one of the main morbidity and mortality factors in the general population and especially in elderly patients. Thus, at the European level, the prevalence of heart failure is 1% in people under 55 years of age but increases to over 10% in people over 70 years of age [1].

With life expectancy increasing to an average of 73 years, the geriatric population is becoming increasingly representative, leading to an increasing incidence of heart failure. In recent years, in developed countries, heart failure has become a growing public health problem, generating higher costs for the diagnosis, treatment and follow-up of these patients [2,3,4,5,6].

Advances in medicine, easier access to medical and care services have led to an increase in the elderly population and consequently its share of heart failure cases. More than a quarter of heart failure patients are over 80 years of age, and among decompensated cases requiring emergency hospitalization, about 1 in 7 are over 80 [7,8,9,10,11,12,13]. As you can see in Figure 1, the predictions are that heart failure in the elderly will more than double by 2040 and triple by 2060 (heart failure in the elderly is set to triple by 2060, according to new data from the Age, Gene/Environment Susceptibility (AGES)—Reykjavík study presented at ESC Congress 2016) [5,6,14,15,16].

Age plays an important role in the deterioration of cardiovascular function, thus the risk of cardiovascular disease among the geriatric population is very high. The prevalence of cardiovascular disease increases with age [5,6,16,17,18]. For example, The American Heart Association reported an incidence of heart disease of 40% at ages between 40–59 years, increasing to 75% at the ages of 60–79 years and reaching 86% for people over 80 years old [19,20,21]. All over the world, the geriatric population poses a great burden on health care systems because of the high prevalence of cardiovascular disease and especially heart failure among them. This burden, reflected in higher costs of treatment, hospitalization and rehabilitation, is directly correlated with the increase in morbidity, frailty and mortality among the elderly. Taking these facts into consideration and the prediction of the rise of the elderly population by 2–3 times until the year 2050 [1,14], the necessity of better understanding the etiology and physiopathology of cardiovascular disease is essential [7,8,9,10,11,12,13,22].

## 2. Objectives

-To analyze the role of brain natriuretic peptides in the diagnosis of heart failure in the elderly and in the differential diagnosis between cardiac failure and respiratory failure.-To understand the way age and other factors influence BNP/NTproBNP levels in geriatric patients, and how this affects the specificity and sensibility of these biomarkers in identifying heart failure in this category of patients.-To analyze the correlation between natriuretic peptides and myocardial left ventricular dysfunction.

This is a systematic review of the most relevant clinical studies and trials conducted in the last 20 years addressing the topic of heart failure in the elderly population. The patients included in these studies were over 65 years old, with a mean age of 75 years, and they were diagnosed with heart failure according to the current guidelines of medical practice. Clinical and imaging data and natriuretic peptides (BNP, NTproBNP) were obtained for all patients.

The majority of these trials were relatively small, retrospective and registry-based trials, including a mean number of 1500 patients per trial, each study reflecting the experience of one medical center (see Table 1).

## 3. Results

Several clinical trials such as “Study of the Role of Plasma NTproBNP in the Diagnosis of Heart Failure” by Bahaar Athavale et al., “Use of NTproBNP Testing to Guide Heart Failure Therapy in the Outpatient Setting”, and “Long-Term Results of Intensified, N-Terminal-Pro-B-Type Natriuretic Peptide-Guided Versus Symptom—Guided Treatment in Elderly Patients with Heart Failure-2013” showed that the NTproBNP is a hormone produced by the heart and that it is a strong prognostic factor for adverse events, hospitalization and mortality in heart failure patients [14,19,23,24,25,26,27,28,29,30,31,32,33,34,35,36,37,38]. Drugs that are proven in clinical trials to be beneficial in heart failure, such as beta blockers and angiotensin converting enzyme inhibitors, also show a reduction of serum levels of NTproBNP [19,20,21,34,35]. Additionally, these studies suggest that NTproBNP could identify patients at high risk of adverse events even in the setting of compensated heart failure.

Natriuretic peptides are also a good predictor of short-term mortality in patients with acute decompensated heart failure. Comparing BNP to NTproBNP, the latter has a longer biological half time than BNP, thus being a more stable molecule with fewer fluctuations. These characteristics make NTproBNP a better marker for screening of heart failure [5,6,14,15,16,17,18,23,24,25,39,40,41].

The first clinical trial that investigated the use of brain natriuretic peptide serum levels as a marker of heart failure was conducted by Mark Richards in the Cardio-Endocrine Research Group. In a population of 205 patients, it was demonstrated that NTproBNP is as useful as BNP in the diagnosis of heart failure. At that time, the standard biomarker of heart failure was BNP, but in recent years the N-terminal fragment of BNP (NTproBNP) was recognized as an important indicator of heart failure [14,23,24,42,43,44]. It was shown that the serum level of NTproBNP was higher among patients with decompensated heart failure and also that it was useful for the diagnosis of the so-called “hidden” heart failure, when heart failure is masked by a concomitant pulmonary disease. NTproBNP was better in making the correct diagnosis of heart failure than medical history, physical examination and other blood tests [2,3,4,7,8,9,10,11,12,45].

Another research study, the PRIDE trial, including 600 patients, showed that NTproBNP was a good marker for the diagnosis of heart failure and also for ruling out heart failure in the emergency department [46].

Recent medical practice guidelines support the use of B-type natriuretic peptide in the diagnosis, prognosis and determination of the severity of heart failure, as well as being useful in guiding treatment [1,5,6,15,16,17,18,41].

In the latest European Society of Cardiology practice guidelines on heart failure (2021), there is a class I level B evidence indication for the use of BNP/NTproBNP as a diagnostic tool. Plasma levels of natriuretic peptides are recommended as a first-line diagnostic test in patients suspected of heart failure for differential diagnosis. An elevated level of natriuretic peptides supports the diagnosis of heart failure and may also guide the investigation algorithm and prognostication [13,14,22,23,24,40,41,47]. However, several factors influence levels of natriuretic peptides, and because of this, their specificity is reduced. Several studies investigated the cut-off values for BNP/NTproBNP in the acute and chronic phases of heart failure. Plasma concentrations of BNP below 35 pg/mL and of NTproBNP below 125 pg/mL make the diagnosis of heart failure highly unlikely [1,15,16,22,40,41,47].

In elderly patients, serum BNP levels may be influenced by a variety of factors in addition to heart failure, with clinical studies showing an increase with age in serum BNP values. Because of the higher basal BNP levels in elderly patients, its negative predictive value in detecting heart failure is lower in the geriatric population.

In an analysis of the Pianoro study, which included patients over 65 years of age and followed parameters that may correlate with mortality and their biological age, it was found that BNP and in particular the precursor NTproBNP correlates best with mortality [1,5,6,16,17,18,36,37,38,48,49,50].

Figure 2 shows the linear relationship between patients’ chronological age and BNP values (BNP age).

Mortality increased from BNP age values above 69 years and peaked at values above 85 years (from 7.9% to 22.8%). Correlating these values with the chronological age of the patients, mortality increased starting from a difference of at least 11 years between chronological age and BNP age [49,50].

Several clinical studies have shown the importance of changes in the dynamics of serum natriuretic peptide levels. Thus, a study published in the Journal of the American College of Cardiology in 2010 included about 2900 patients over 75 years of age, measuring baseline BNP levels at the time of inclusion and at 2–3-year intervals. The results of the study showed an increased incidence of heart failure and cardiovascular mortality with increasing NTproBNP values over time. Cardiovascular risk began to increase from serum NTproBNP values of 190 pg/mL. Patients who had an increase of more than 25% in natriuretic peptide values had the highest risk of heart failure and cardiovascular mortality, regardless of baseline natriuretic peptide levels. What this study demonstrates is that serum NTproBNp values are in a continuous dynamic, and this is reflected in cardiovascular risk [15].

In a study led by Passantino and colleagues, published in the Journal of the American Geriatrics Society, a cohort of elderly patients hospitalized for heart failure was analyzed, identifying prognostic factors associated with short- and long-term mortality [36]. This retrospective study included 279 patients with a mean age of 80 years and demonstrated the importance of NTproBNP natriuretic peptide as an independent prognostic factor for all-cause mortality in geriatric patients admitted for heart failure. Reported mortality was 36% at 1 year and 77% at 5 years, but it is difficult to differentiate the exact cause of death for each individual because of the multiple comorbidities present. Non-cardiovascular and cardiovascular mortality rates varied according to BNP values, so that a cut-off BNP value that predicted mortality at 2 months and 1 year was 8200 pg/mL [36].

Despite the positive predictive value of BNP for mortality in elderly patients with heart failure, intensification of treatment in high-risk patients based on increased serum natriuretic peptide values, although resulting in improved symptomatology and quality of life, had no influence on mortality. This is probably also due to the high proportion of non-cardiovascular mortality among elderly patients due to multiple associated pathologies [7,8,9,10,11,12,13,15,16,22,40,41,45].

Several clinical trials have evaluated the efficacy of heart failure therapy guided by serum BNP/NTproBNP values, but without favorable results [19,20,21,35,51,52,53,54]. Thus, the GUIDE-IT trial was stopped after only 18 months due to the lack of benefit of BNP-guided therapy [54]. The study included a total of 894 patients diagnosed with heart failure, of which 446 patients were treated according to a natriuretic peptide-guided strategy and the rest of the patients were treated with conventional therapy. For patients with guided BNP therapy, treatment was titrated to achieve a target NTproBNP value below 1000 pg/mL.

In an article published in the European Heart Journal by Richard W. Troughton and co-workers, a meta-analysis of BNP-guided therapy in patients with heart failure was proposed. The data were compared and analyzed according to various criteria, including age, with patients being divided into two age categories: under 75 and over 75. It was found that in patients under 75 years of age, NTproBNP-guided therapy led to a decrease in mortality and hospitalization rates, benefits that were not found in patients over 75 years of age [48].

Its previous meta-analyses also suggested a possible 20–30% reduction in all-cause mortality when using natriuretic peptide-guided heart failure therapy.

However, the European Society of Cardiology considered the evidence insufficient for a firm recommendation of BNP-guided therapy in heart failure [1].

In the BED (BNP Usefulness In Elderly Dyspnoeic Patients) study published by Plichart et al. in the European Journal of Heart Failure [24], 383 elderly patients, aged 80 years or older, hospitalized for dyspnea, were measured for acute atrial natriuretic peptide. On the basis of the medical investigations, the patients were classified, independently of the BNP value, into two categories: cardiac dyspnea and respiratory dyspnea. Subsequently, clinical data were correlated with BNP values, which proved not to improve the discrimination between cardiac or respiratory etiology of dyspnea. In this study, the sensitivity for the diagnosis of heart failure was 90% for a BNP value of 100 ng/L, and a specificity of 90% was achieved at BNP values of 800 ng/mL [24].

In the HFinCH study, led by James Mason et al., published in 2013 [55], 405 elderly people (aged 65 years or older; mean age 84 years) were screened for heart failure. Clinical data, ECG and echocardiography were analyzed and the diagnosis of heart failure was made according to European guidelines for medical practice. Subsequently, the patients were sampled for BNP and NTproBNP. Patients diagnosed with heart failure were separated into two categories: patients with left ventricular systolic dysfunction (reduced LVEF) and patients with preserved LVEF. Subsequently, BNP/NTproBNP values were compared between patients with heart failure and patients without. BNP values were lower in patients without heart failure compared to those with heart failure. Among patients diagnosed with heart failure and left ventricular systolic dysfunction, BNP and NTproBNP values were associated with disease severity as follows: mean BNP values of 270 pg/mL for mild forms, 680 pg/mL for medium forms and 428 pg/mL in severe forms of the disease. BNP had a threshold value for the detection of patients with left ventricular systolic dysfunction of 145 pg/mL with a sensitivity of 76% and specificity of 75%, and for NTproBNP the threshold value was 1000 pg/mL with a sensitivity of 73% and specificity of 76%. In contrast, for heart failure patients with preserved left ventricular function, BNP and NTproBNP have no diagnostic utility. The analysis of the data obtained from the study showed that both BNP and NTproBNP can be used in the diagnosis of heart failure with left ventricular systolic dysfunction with a sensitivity of only 76%, so one in four patients would remain undiagnosed based on these biomarkers alone [55].

The Breathing Not Properly study [32], was a multinational study involving 1586 patients presenting to emergency departments for dyspnea who had their serum BNP measured at presentation. BNP was a stronger predictor of congestive heart failure in younger patients than in older patients [32]. This was because serum BNP values increase with age, thus reducing its specificity in diagnosing heart failure.

In a smaller study, on a cohort of 64 patients with a mean age of 84 years, a cut-off value for BNP of <129 ng/L showed a 90% negative predictive value in excluding heart failure [35].

In another study of 201 patients admitted for dyspnea, 45% of whom were over 75 years of age, BNP and NTproBNP values played an important role in discriminating acute heart failure from other conditions. In elderly patients, the cut-off values for BNP were higher than in younger patients, and the specificity of natriuretic peptides decreased with age [56].

In the British Medical Journal in 2018, Kathryn Taylor et al. published a meta-analysis on the diagnostic accuracy of natriuretic peptides for chronic heart failure in ambulatory care. At thresholds >100 pg/mL, BNP had a sensitivity of 0.95 reduced between 0.46 and 0.97 and a specificity between 0.31 and 0.98 at a threshold <100 pg/mL. For NTproBNP, the specificity was 0.99 and the specificity was 0.60 at a threshold >136 pg/mL. There was no significant difference in diagnostic accuracy between BNP and NTproBNP [49].

More recently, in 2021, Nevis et al. published the study Use of B-Type Natriuretic Peptide (BNP) and N-Terminal proBNP (NT-proBNP) as Diagnostic Tests in Adults With Suspected Heart Failure: A Health Technology Assessment, to evaluate the diagnostic accuracy, clinical impact and cost-effectiveness of BNP/NTproBNP testing in heart failure. Natriuretic peptides had a high sensitivity (80–96%) and low likelihood value [37]. In one review, the use of natriuretic peptides in the diagnosis of heart failure in the emergency department decreased the mean length of the hospital stay [37]. However, they did not reduce 30-day hospital readmission rates or hospital mortality rates. They also showed the cost-effectiveness of the use of natriuretic peptides testing in addition to standard clinical investigations for heart failure.

The mean values for the specificity of BNP/NTproBNP in the diagnosis of heart failure are 80–85% and the mean sensibility is 75% (with NTproBNP being a stronger marker of heart failure), but there was a great variation of cut-off values between these trials [1,18,20,21,36,37,38,42,43,44,48,49,50,51,52,53,54,55,56,57].

Each study attempted to find cut-off values for BNP/NTproBNP that discriminated the cardiac and respiratory etiology of dyspnea in geriatric patients. At present, there is no general consensus on these cut-off values. Although a high specificity has been achieved at higher BNP values, no cut-off BNP value has been found that presents a sufficient probability for a definite diagnosis of cardiac insufficiency.

Factors underlying the reduced specificity of natriuretic peptides in geriatric patients are age, atrial fibrillation, high prevalence of preserved left ventricular systolic function in elderly patients, renal dysfunction, chronic lung disease and other comorbidities.

Additionally, in geriatric patients, the diagnosis of heart failure is more difficult because of the multiple comorbidities present, which interfere with the diagnosis. Elderly patients are more likely to be less investigated in emergency departments, mainly because of the severity of symptoms at presentation, so their classification into a particular diagnosis is more difficult.

In elderly patients, unlike younger patients, BNP does not yet discriminate between the cardiac or respiratory origin of dyspnea as the main form of heart failure see Figure 3 below.

More clinical trials including geriatric patients are needed to establish the exact role of natriuretic peptides in the diagnosis of heart failure.

## 4. Discussions

Elderly patients are under-represented in clinical trials on the diagnosis and treatment of heart failure. These studies underlie the development of treatment guidelines and form the foundation of modern evidence-based medicine. On average, less than 30% of the patients selected in these clinical trials were over 70 years of age. Many of these patients are excluded from clinical trials because of comorbidities.

The particularities of the elderly patient, which make the management of heart failure difficult, are the presence of comorbidities, frailty, cognitive impairment and polypharmacy [7,8,9,10,11,12,50,55,58].

Approximately 60% of elderly patients with heart failure have at least three comorbidities. Among these comorbidities, the most common are hypertension and heart rhythm disorders, and among non-cardiac conditions these include chronic renal failure, diabetes mellitus and anemia [2]. All these comorbidities make the geriatric patient a complex case, which has an impact on the diagnosis and treatment algorithm of heart failure in these patients.

Although the benefits of standard therapies in heart failure, as evidenced by subgroup analyses in clinical trials, are present in the elderly population, these drugs have been found to still be underutilized [1,14,24,39]. This is mainly due to contraindications as well as the risk of adverse reactions. Elderly patients are more prone to adverse reactions, being more vulnerable due to comorbidities and frailty. The TIME-CHF trial (Trial of Intensified versus Standard Medical Therapy in Elderly Patients with Congestive Heart Failure) studied the titration of specific medication according to serum natriuretic peptide levels in patients over 60 years of age, without demonstrating benefits in patients over 75 years of age [57]. However, geriatric patients receive suboptimal doses of medication. On the other hand, the optimal doses of drugs that are proven by clinical trials to improve the survival of patients with heart failure are generally chosen by individual clinicians according to the patient’s symptoms and clinical signs. These subjective criteria often lead to medication under-dosing, so target doses are rarely reached. Hence the need for objective indices to guide heart failure therapy.

In addition to drug treatment, heart failure requires invasive interventions on the heart, such as coronary angiography followed by possible coronary revascularization by percutaneous angioplasty or coronary artery bypass grafting, pacemaker implantation, intramyocardial resynchronization devices, intracardiac defibrillator and radiofrequency ablation for heart rhythm disorders. All of these therapies lead to improved quality of life and survival, but the decision to perform them on an individual geriatric patient requires a thorough assessment. This requires multidisciplinary teams of cardiologists, neurologists, geriatricians, psychiatrists, etc. Assessment of comorbidities, frailty, life expectancy, risk/benefit ratio, etc., must be taken into account [5,6,15,16,17,18,41]. Although these interventions have proven beneficial effects, European registries show an under-use of them in elderly patients, so that about 30% of intramyocardial resynchronization devices have been implanted in patients over 75 years of age, and a Spanish registry shows a 15% share of patients over 75 years of age in intracardiac defibrillator implantation [14]. Again, objective indices (biomarkers) are needed to stratify prognosis and cardiovascular risk in order to guide the medical team in making an appropriate therapeutic decision for each individual patient.

The main symptoms of heart failure, such as exercise intolerance and dyspnea, can also be interpreted in the context of older age and the functional disability of geriatric patients, which is why the diagnosis of heart failure may be overlooked.

In a study of the distribution of risk factors for the development of heart failure according to age, a higher proportion of known classical risk factors was found in the younger population compared to the elderly. In other words, traditional predictors of heart failure were less common in elderly patients. For example, the presence of smoking as a risk factor was 21% in younger patients versus 13% in older patients and diabetes was present in 14% of younger patients versus 7% of geriatric patients [26,33,34].

Under these conditions, there is a growing interest in developing effective methods for the diagnosis, screening, prognosis and treatment guidance of heart failure. These methods need to be as specific and sensitive as possible to be clinically relevant.

Although the scientific data on the etiology and pathophysiology of heart failure are substantial, the imaging methods used are increasingly more powerful with high sensitivity, and diagnostic criteria are constantly being revised in guidelines for medical practice.

The concept of biochemical tests, using various biomarkers such as natriuretic peptides and troponin, is increasingly taking shape in current clinical trials.

The diagnosis of heart failure is based on the patient’s clinical history, the physical examination of the patient performed by the doctor and the results of the complementary imaging tests: chest radiography and echocardiography. About 30–50% of patients presenting in emergency departments for decompensated heart failure are misdiagnosed, because of the atypical manifestations, lack of imagery and the clinical signs that are not always obvious [32]. Thus, there is a necessity for new diagnostic tests to help clinicians in the diagnosis of heart failure and make a better differential diagnosis with other diseases with a similar manifestation as heart failure. The correct and timely diagnosis of heart failure would make the initiation of adequate treatment possible, and by doing so would increase the results of the treatment, reduce the hospitalization duration, reduce the costs of therapy and improve prognosis of the patients.

Diagnosis of heart failure, especially in elderly patients, is difficult because symptoms such as fatigue and dyspnea on exertion, as well as clinical signs such as swollen jugular veins or tibial edema, are not specific to heart failure, but may be caused by a range of comorbidities. Differential diagnosis between cardiovascular disease and other pathologies is difficult in geriatric patients.

Additionally, the clinical symptoms and signs of heart failure often become apparent in the later stages of the disease, so diagnosis is made late and the therapeutic benefits are greatly reduced.

As for dyspnea, the most common complaint of patients with heart failure, studies show uncertainty about its etiology. For example, the “Breathing not Properly” study reports diagnostic errors in about half of the cases presented for dyspnea. The difficulty of diagnosing heart failure is even greater in patients presenting for the first time with symptoms of dyspnea. Clinical examination of the patient is often not sufficient in developing an accurate diagnosis, and imaging methods such as chest radiography have neither the necessary sensitivity nor specificity. Echocardiography can detect changes in left ventricular ejection fraction, but for heart failure with preserved ejection fraction, it is not as useful.

The need for complementary methods (biomarkers) for differential and early diagnosis of heart failure is becoming more and more evident, even in its subclinical stages. These methods need to have increased specificity and sensitivity and be widely available.

Natriuretic peptides, in particular B-type natriuretic peptide (BNP) and its fraction NTproBNP, have gained in recent years an increasingly important role in the screening, diagnosis and treatment of heart failure.

The B-type natriuretic peptide is released by myocardial cells in response to increased parietal stress under the conditions of volume or intramyocardial pressure overload that occur in heart failure. BNP has a natriuretic and peripheral vasodilator effect, thus reducing this intramyocardial overload.

Geriatric patients show an increased prevalence of heart failure with preserved left ventricular ejection fraction, which makes diagnosis more difficult, especially in the presence of comorbidities and predominant symptoms, such as dyspnea on exertion and reduced functional capacity, which is often attributed to simple ageing. Natriuretic peptides could be useful in the diagnosis of heart failure for this type of patient, but the interpretation of their serum values may be influenced by several factors such as chronic kidney disease, sepsis, hyperthyroidism, chronic anemia, atrial fibrillation and others. All these comorbidities, very common in elderly patients, modifying basal natriuretic peptide values independently of the presence of heart failure lead to the need to establish different cut-off values of these biomarkers in geriatric patients. Age and chronic kidney disease appear to be the major determinants of serum BNP/NTproBNP values. It was also found that even patients without a diagnosis of heart failure, but with elevated serum natriuretic peptide values, showed increased cardiovascular events and mortality. This raises the possibility of using natriuretic peptides to screen for silent heart disease in elderly patients [15,47].

## 5. Conclusions

Most clinical studies have shown that natriuretic peptides have low discriminatory power for the diagnosis of heart failure in elderly patients. The importance of these peptides is still the subject of debate and research, and more clinical studies are needed to provide data on their usefulness in the diagnosis, treatment and prognosis of heart failure in elderly patients. In general, natriuretic peptides have been more sensitive in diagnosing forms of heart failure with left ventricular systolic dysfunction and less sensitive in forms with preserved systolic function.

## Figures and Tables

**Figure 1 diagnostics-13-01512-f001:**
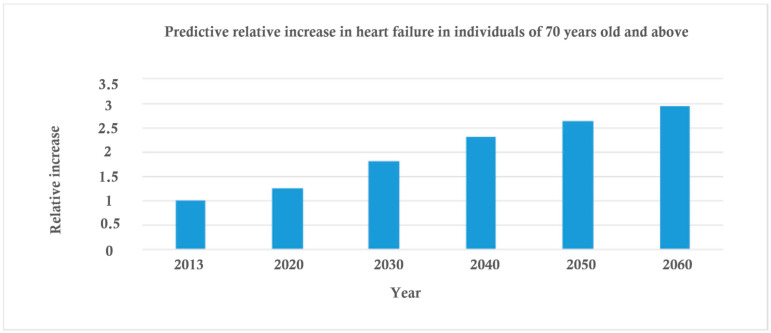
Predictive relative increase in heart failure in ≥70 years old individuals, according to data from the Age, Gene/Environment Susceptibility (AGES)—Reykjavík study [5,6,14,15,16].

**Figure 2 diagnostics-13-01512-f002:**
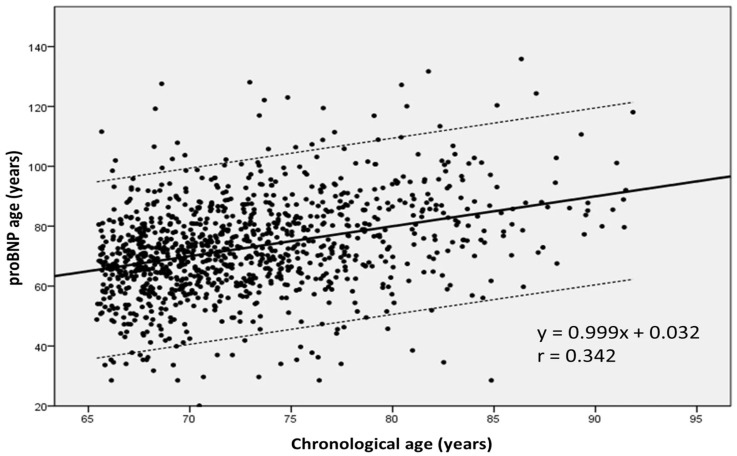
A linear relationship between patients’ chronological age and BNP values was proven- regression line and 95% confidence interval between chronological age and proBNP age; the equation is: proBNP age = 0.999 × age + 0.032 (r = 0.342, *p* < 0.0001) [49,50].

**Figure 3 diagnostics-13-01512-f003:**
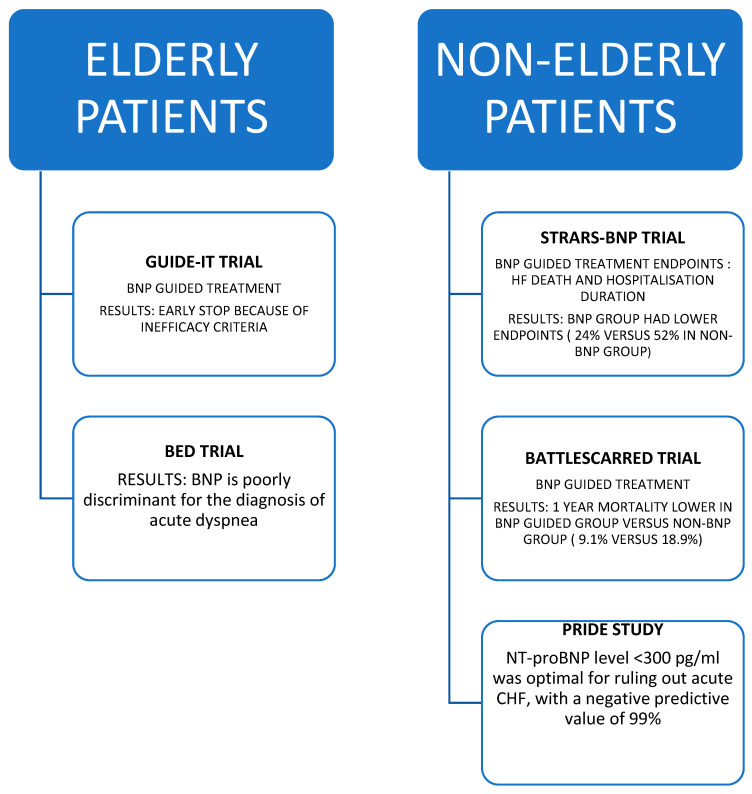
Comparison between elderly and non elderly patients heart failure trials.

**Table 1 diagnostics-13-01512-t001:** Review of relevant trials using BNP for heart failure assessment in elderly.

*Trial*	Year	No Patients	Endpoints	Results
*GUIDE-IT*	2012–2016	894	If natriuretic-based therapy improves clinical outcome in heart failure	Negative results; study ended in 2016
*BED trial*	2013	383	Use of BNP in diagnosis of heart failure in very old patients	BNP is a poor discriminant of heart failure in the elderly
*HFinCH trial*	2009–2010	405	BNP use to rule out heart failure	BNP/NTproBNP miss one in three patients with heart failure
*“Breathing not properly” trial*	2012	-	Use of natriuretic peptides in the diagnosis of acute heart failure, guide therapy, discharge management, prevent readmissions, etc.	
*Predictors of Long-Term Mortality in Older Patients Hospitalized for Acutely Decompensated Heart Failure: Clinical Relevance of Natriuretic Peptides*	2016	279	Long-term mortality	In addition to EF and comorbidities, NT-pro-BNP remained independently prognostic among elderly patients hospitalized with heart failure.

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
