# Peer review of "The Use of Brain Natriuretic Peptide in the Evaluation of Heart Failure in Geriatric Patients"

_diagnostics, 2023, doi:10.3390/diagnostics13091512_

Round 1

Reviewer 1 Report (New Reviewer)

The review is only mildly interesting, mainly because it doesn't seem to focus on anything new on the subject.
the introduction is very long and needs to be rewritten and shortened considerably. there are repeated phrases (such as the reference to fragility).
the materials and methods should be better written. distinguish what is reported in the results from the actual discussion. Discussion that looks a lot like a list of studies. the conclusions are too long and risk not passing on the most important concept.

Minor comments:
page 3: please rewrite hospitalizationof as hospitgalization of
page 6 please write correlation
page 7 ...angiotensin ... suggest
page 8 this phrase "the figure below..." is duplicated please deleted one.
page 14 aetiology of

Author Response

Thank you for your review, insights and suggestions. I have made modification to the introduction and conclusions sections. Also I have restructured the article regarding the discussions and results and have made the minor modifications.

I await your response and should it need more modifications please let me know

Thank you

Dr. Mihai Marinescu

Reviewer 2 Report (New Reviewer)

The manuscript submitted to me for review is a review of clinical studies and trials conducted over the past 20 years on heart failure in the elderly patient population.

The authors identified three study objectives, including:

- evaluation of the role of brain natriuretic peptides in the diagnosis of this disease and in the differential diagnosis between heart failure and respiratory failure;

- clarification of the impact of BNP/NTproBNP levels on the diagnosis of heart failure in this category of patients;

- indication of the correlation between natriuretic peptides and left ventricular myocardial dysfunction.

After a careful reading of the manuscript, I conclude that the abstract, introduction and remaining chapters comprehensively and adequately capture the issues discussed.  The conclusions presented by the authors are consistent with the evidence and address the main research problem. In addition, I believe that the selection of bibliographic items is appropriate.

 I recommend publication in its present form.

Author Response

Thank you for your kind review

Dr Mihai Marinescu

Round 2

Reviewer 1 Report (New Reviewer)

the paper was improved by authors, however I should suggest to add a table summarizing the mail HF trial in elderly population and one figure with differences between HF trials in elderly people vs other HF trial.   

Author Response

Thank you for your second review and your good suggestions

As such I have introduced a table of natriuretic peptides in the elderly and a comparison of the most relevant trials of heart failure in elderly and non-elderly

I kindly await for your review

Thank you

Dr Mihai Marinescu

Round 3

Reviewer 1 Report (New Reviewer)

The paper was sufficiently improved by authors. I have not further comment or suggestion.

This manuscript is a resubmission of an earlier submission. The following is a list of the peer review reports and author responses from that submission.

Round 1

Reviewer 1 Report

Authors are attempting to evaluate the use of natriuretic peptides among HF older patients.

However, I feel that even from the title there is a confusion regarding natriuretic peptides vs BNP vs NTpro BNP.

I personally think that natriuretic peptides are established biomarkers in the diagnosis of HF (according to ESC guidelines, BNP>35 pg/ml, NTprobnp >135 pg/ml). It is known, as the authors mention that the cutoffs for elder people are higher, however, from the studies that are presented, there is no evidence to support a change in paradigm.

Authors could re construct the manuscript using separate sections for the use of natriuretic peptides in elderly for diagnosis vs prediction vs prognosis vs treatment guidance. On the same note, I think introduction is way too long and most of the sections could be used in “discussion”.

It would be nice from the authors to have a table with the reviewed studies and the most important results per section (diagnosis vs prediction vs…..).

In the presented studies there is no clear separation between the natriuretic peptide that was used (BNP vs NTprobnp). Besides the explanation that authors have given regarding he difference these two biomarkers, I think the different effect of ARNI on the levels of these two biomarkers (favoring ntpro) should also be mentioned.

Other Comments

1)     I would recommend deleting the word “brain” from the title

2)     Replace “ at European level” with “in Europe”

3)     Abstract needs rewriting

4)     In introduction the first reference has the number of 50 instead of 1. Citation order  should be updated

5)     Page 1 line 33: replace medical and care with Healthcare

6)     Page 1line 36, replace “ as you can see”

7)     Permissions should be stated for the 1st figure

8)     Page 2 line 48: please rephrase “ the geriatric….health care systems”

9)     Page 2 line 52-53, : why is their need for better understanding of pathophysiology , beyond what we know currently?

10)  Page 2 line 56-57: please rephrase “these studies…medicine”

11)  Page 2 line 70-71, 78: I don’t think that “ the optimal doses…clinical signs “ is accurate. There are guidelines which recommend uptitration to max tolerated doses. Please rephrase.

12)  Page 3 line 82 , please replace “on the heart” with CV interventions

13)  Page 3 line 94-96. I believe that there are already established biomarkers to help with prognosis etc.

14)  Page 3 128: jugular vein distention, is both sensitive and specific for HF.(I would recommend to delete it)

15)  Page 4 lines 139-142. I think that there are guidelines that are utilizing the use of echocardiographic parameters in particular for the diagnosis of HF (ESC/ACC HF guidelines)

16)  Page 4 line 166: replace sensibility with sensitivity

17)  I think the whole manuscript needs restructuring, the sections material, methods, results and discussions are not applicable, since this is mostly a review paper. No statistical analysis, or comparisons were performed.

Reviewer 2 Report

Reviewing the manuscript entitled, “The use of Brain Natriuretic Peptide in the evaluation of Heart Failure in Geriatric Patients” by Marinescu M et al., this is a review manuscript about heart failure markers in in the elderly patients. Although this is an important review, there are major concerns as follows.

The introduction section is too long. The purpose of this review is whether BNP or NT-pro BNP is useful as a biomarker for heart failure symptoms in the elderly, as described in the abstract. Nevertheless, the introduction section is all about explaining the symptoms and etiology of heart failure in the elderly, and quoting such information is sufficient. In addition, the secretion mechanism of BNP and NT-pro BNP from the myocardium, the difference between the two, and the relationship with drugs as ARNI are not described at all. This is clearly a poor introduction to this review title.

 In 2. Materials and methods, the authors should establish a table for the EBMs covered in this review. There is no mention of the eligibility of the EBMs selected. If the relationship between BNP and ARNI is not clearly described, the content may be blurred.

 The authors should describe that the difference between bioactive BNP, which is constitutively secreted mainly from myocardial cells, and its inactivated fragment, NTproBNP, the relationship with drugs, and their respective metabolic pathways. Since this is a review manuscript, you should explain them using figures if possible.

You mentioned description of elderly comorbidities with the heart failure. NTproBNP increases in renal insufficiency.

 There is no figure legend attachment. The authors should modify them. 

 ARNI demonstrated efficacy exceeding that of RAS inhibitors in the treatment of heart failure on PRADIGM-HF in 2014, and since then ARNI has been used as the first line of heart failure in many countries. ARNI is a complex composed of ARB and Neprilysin inhibitor at a ratio of 1:1, and Neprilysin inhibitor is a heart failure therapy aimed at increasing BNP/ANP/CNP. Therefore BNP, which is pharmacologically elevated by ARNI, is no longer a heart failure marker when this is used. Moreover, as mentioned above, the therapeutic effect of ARNI beyond that of RAS inhibition, and there is a clear problem with the argument that BNP is used as a heart failure marker at this stage.